# Volumetric Analysis of Glioblastoma for Determining Which CpG Sites Should Be Tested by Pyrosequencing to Predict Temozolomide Efficacy

**DOI:** 10.3390/biom12101379

**Published:** 2022-09-26

**Authors:** Tomohiro Hosoya, Masamichi Takahashi, Calvin Davey, Jun Sese, Mai Honda-Kitahara, Yasuji Miyakita, Makoto Ohno, Shunsuke Yanagisawa, Takaki Omura, Daisuke Kawauchi, Yukie Ozeki, Miyu Kikuchi, Tomoyuki Nakano, Akihiko Yoshida, Hiroshi Igaki, Yuko Matsushita, Koichi Ichimura, Yoshitaka Narita

**Affiliations:** 1Department of Neurosurgery and Neuro-Oncology, National Cancer Center Hospital, 5-1-1 Tsukiji, Chuo-ku, Tokyo 104-0045, Japan; 2Humanome Laboratory, 2-4-10 Tsukiji, Chuo-ku, Tokyo 104-0045, Japan; 3Department of Pathology, National Cancer Center Hospital, 5-1-1 Tsukiji, Chuo-ku, Tokyo 104-0045, Japan; 4Department of Radiation Oncology, National Cancer Center Hospital, 5-1-1 Tsukiji, Chuo-ku, Tokyo 104-0045, Japan; 5Department of Brain Disease Translational Research, Juntendo University Faculty of Medicine, 2-1-1 Hongo, Bunkyo-ku, Tokyo 113-8421, Japan

**Keywords:** glioblastoma, volumetric analysis, pyrosequencing, CpG site

## Abstract

The aim of the present study was to determine which individual or combined CpG sites among O^6^-methylguanine DNA methyltransferase CpG 74–89 in glioblastoma mainly affects the response to temozolomide resulting from CpG methylation using statistical analyses focused on the tumor volume ratio (TVR). We retrospectively examined 44 patients who had postoperative volumetrically measurable residual tumor tissue and received adjuvant temozolomide therapy for at least 6 months after initial chemoradiotherapy. TVR was defined as the tumor volume 6 months after the initial chemoradiotherapy divided by that before the start of chemoradiotherapy. Predictive values for TVR as a response to adjuvant therapy were compared among the averaged methylation percentages of individual or combined CpGs using the receiver operating characteristic curve. Our data revealed that combined CpG 78 and 79 showed a high area under the curve (AUC) and a positive likelihood ratio and that combined CpG 76–79 showed the highest AUC among all combinations. AUCs of consecutive CpG combinations tended to be higher for CpG 74–82 in exon 1 than for CpG 83–89 in intron 1. In conclusion, the methylation status at CpG sites in exon 1 was strongly associated with TVR reduction in glioblastoma.

## 1. Introduction

Glioblastoma is the most common and aggressive malignant brain tumor in adults. The standard treatment for newly diagnosed glioblastoma is maximal safe resection and postoperative local radiotherapy with concomitant temozolomide, followed by adjuvant temozolomide [1,2]. However, the median and 2-year survival rates with this standard treatment are 14.6 months and 26.5%, respectively [2].

O⁶-methylguanine DNA methyltransferase (*MGMT*) promoter methylation is a prognostic factor for glioblastoma. *MGMT* encodes a DNA repair enzyme that removes alkyl adducts from the O^6^ position of guanine in tumor DNA, which is damaged by alkylating agents, such as temozolomide, and converts the guanine in tumor DNA to O^6^-methylguanine, exerting antitumor effects by impairing DNA replication. Therefore, *MGMT* expressed in tumor cells removes the guanine alkyl group and attenuates its antitumor effects [3]. Conversely, patients with *MGMT* promoter methylation respond better to temozolomide and have a better prognosis owing to MGMT protein suppression. Thus, the *MGMT* promoter methylation status is a prognostic and predictive marker for temozolomide in glioblastomas, and its evaluation helps predict the treatment efficacy and prognosis. 

Pyrosequencing is the gold standard test for *MGMT* promoter methylation in glioblastoma. It can analyze several CpG positions simultaneously and generate quantitative results for each analyzed CpG position individually, with rapid parallel processing of numerous samples [4]. Using pyrosequencing, Malley et al. revealed that the methylation status of CpG 73–90 (differentially methylated region 2 (DMR2)) among 98 CpG sites in the *MGMT* promoter region correlated strongly with MGMT protein suppression [5]. In addition, pyrosequencing has been used to analyze the *MGMT* methylation percentage in DMR2 and determine the MGMT protein expression; however, the CpG sites that should be tested with pyrosequencing have not been determined. The methylation level at each CpG site correlates with the *MGMT* messenger RNA (mRNA) expression [6,7] and survival [8] in patients with glioblastoma. However, the *MGMT* promoter methylation status does not always correlate with the *MGMT* mRNA expression or survival. The *MGMT* mRNA expression is inconsistent with the *MGMT* promoter methylation status in at least 15% of cases [6], probably because of the cutoff point determining the *MGMT* promoter methylation status or mRNA expression and contamination of normal brain tissue. In addition, survival data vary depending on the population or sample. As aforementioned, no method can accurately detect the MGMT protein expression; therefore, CpG sites that should be tested with pyrosequencing should be explored from various perspectives, including observation in clinical practice.

*MGMT* promoter methylation percentage (*MGMT*pm%) obtained with pyrosequencing correlates with the tumor volume ratio (TVR) obtained with magnetic resonance imaging (MRI) in isocitrate dehydrogenase (IDH)-wildtype glioblastoma [9]. The aim of the present study was to determine which individual or combined CpGs mainly affect the response to chemotherapy resulting from *MGMT* CpG methylation using statistical analyses focused on TVR.

## 2. Materials and Methods

### 2.1. Patients

In this retrospective study, we enrolled patients with newly diagnosed IDH wildtype glioblastoma who underwent surgery followed by local radiotherapy equivalent to 60 Gy and concomitant chemotherapy with temozolomide [2]. Local radiotherapy (60 Gy in 30 fractions) and chemotherapy were simultaneously administered within 1 month of surgery for all patients. The study protocol was approved by the Internal Review Board of the National Cancer Center Hospital. Of the 350 patients with primary glioblastoma who underwent surgery and received temozolomide-based chemoradiotherapy at the National Cancer Center Hospital (5-1-1, Tsukiji, Chuo-ku, Tokyo, Japan) between September 2006 and December 2021, we enrolled those who had postoperative volumetrically measurable residual tumor tissue and received adjuvant temozolomide therapy for at least 6 months after the initial chemoradiotherapy. Exclusion criteria were: no measurable contrast-enhanced lesions after gross total resection; current bevacizumab, nivolumab, procarbazine, and novoTTF therapy in addition to temozolomide; and reoperation for tumor recurrence within 6 months. The clinical characteristics of each patient were examined. 

### 2.2. TVR

The volume of contrast-enhancing lesions was calculated using 1.5T–3T MRI scans acquired before and 6 months after the initial chemoradiotherapy. The volume was calculated by multiplying the gross area of contrast-enhanced lesions in each section by the slice thickness. Non-contrast-enhanced areas within the contrast-enhanced areas were measured as lesions, but obvious cystic lesions or postoperative extraction cavities were excluded. TVR was defined as the tumor volume 6 months after the initial chemoradiotherapy divided by that before the start of chemoradiotherapy [9]. 

### 2.3. Molecular Analysis

DNA was extracted from frozen tumor or formalin-fixed paraffin-embedded (FFPE) tissues using DNeasy Blood & Tissue Kits (Qiagen, Tokyo, Japan), and bisulfite modification of 500 ng genomic DNA was performed using EZ DNA Methylation Kits (Zymo Research, Orange, CA, USA). Pyrosequencing primers were designed to cover 16 CpG sites (CpG 74–89) of the *MGMT* promoter [10]. Pyrosequencing of IDH1/2 and *MGMT* promoter was performed using PyroGold Q96 SQA Reagents and PyroMark Q96 software (version 2.5.7) in a pyrosequencing96 pyrosequencer (Qiagen, Tokyo, Japan), according to the manufacturer’s instructions. Data were analyzed using PyroMark Q96 software, as described previously [10,11]. *MGMT*pm% was calculated by averaging 16 CpG islands (74–89) and analyzed using the pyrosequencing method, as described previously, with some modifications [10].

### 2.4. Statistical Analysis

JMP version 14 was used for all statistical analyses, and statistical significance was set at *p* < 0.05. The volumetric assessment was performed 6 months after the start of chemoradiotherapy, based on the Response Assessment in Neuro-Oncology (RANO) criteria [12]. Volumetric complete response (CR), partial response (PR), stable disease (SD), and progressive disease (PD) were defined as no residual tumor, ≥50% decrease, <50% decrease to <25% increase, and ≥25% increase in TVR, respectively. Multiple lesions were evaluated based on the total volume of each lesion. The patients were stratified into two groups: CR/PR and SD/PD groups, using a TVR cutoff of 0.5 (50% volumetric decrease or not) according to the RANO criteria at 6 months after the initial chemoradiotherapy. The receiver operating characteristic (ROC) curve was analyzed to evaluate predictive values for TVR as a response to adjuvant therapy among averaged methylation percentages of individual or combined CpGs. We chose all individual CpGs and two, three, four, and five consecutive CpG combinations for the ROC curve analysis and compared the positive likelihood ratio (LR+) and the area under the curve (AUC) as indices of predictive accuracy. LR+ was calculated using the following formula: LR+ = sensitivity/1 − specificity. It quantifies the increase in the probability of an event (50% or more volumetric reduction) when the test is positive. A higher LR+ is considered to have a greater value for diagnostic tests [13].

## 3. Results

### 3.1. Baseline Patient Characteristics

Table 1 shows the baseline patient characteristics before the start of chemoradiotherapy. We enrolled 44 patients, including 22 (50.0%) men and 22 (50.0%) women, with a median age of 65 years (interquartile range (IQR), 55.3–73.8) (Table 1). Twenty-three (52.3%) patients were aged 65 years or older, and 21 (47.7%) patients were younger than 65 years. Karnofsky Performance Scale scores ≥80 and <80 were 59.1% and 41.0%, respectively. The extent of resection was categorized into 90–99% and <90% tumor removal, with 19 (43.2%) and 25 (56.8%) patients, respectively. The median Ki-67 staining index of each tumor was 28.3% (IQR, 16.5–49.8%). IDH wildtype was confirmed in all the patients. The *MGMT* promoter was examined in 39 frozen samples and 5 FFPE samples from 44 patients, and the median *MGMT*pm% was 5.5% (IQR, 0.8–31.6%). The heatmap shows correlation coefficients between every two CpG sites (Figure 1). CpG sites within the range of CpG 79 to CpG 83 correlated strongly.

The Ki-67 staining index and methylation percentages of CpG 74–82 and 83–89 were stratified by the median Ki-67 index and the cutoff value of 9%, respectively, as reported previously. IDH 1 and 2 mutation was not detected in all the patients participated.

### 3.2. MGMTpm% and TVR at All CpG Sites

Considering that temozolomide therapy efficacy is mainly influenced by the *MGMT* promoter methylation status, the ROC curve was analyzed to evaluate predictive values of TVR in response to temozolomide among the averaged methylation percentages of individual or combined CpG sites. Among all individual CpGs and two, three, four, and five consecutive CpG combinations, AUC and LR+ were the highest for combined CpG 78 and 79 in the ROC curve analysis (Table 2). AUCs of consecutive CpG combinations were higher for CpG 74–82 than for CpG 83–89 (Figure 2). Combined CpG 76–79 showed the statistically highest AUC among all CpG combinations and showed significantly (*p* = 0.0283 in the DeLong test) higher AUC than that of combined CpG 84–87, which showed the lowest AUC among all four consecutive CpG combinations.

## 4. Discussion

In the present study, combinations including CpG 78 and 79, particularly CpG 76–79, showed the statistically highest AUC and LR+. AUCs of consecutive CpG combinations tended to be higher for CpG 74–82 than for CpG 83–89. Since CpG 74–82 is located in exon 1, and CpG 83–89 is located in intron 1 [14], we hypothesized that the methylation status at CpG sites in exon 1 mainly contributes to MGMT activity and response to chemoradiotherapy. 

Since patients with glioblastoma without *MGMT* promoter methylation are resistant to temozolomide and have a poor prognosis, the treatment strategy depends on the *MGMT* promoter methylation status. Although the usefulness of the *MGMT* promoter methylation analysis in patients with glioblastoma for predicting the response to chemoradiotherapy with temozolomide and the prognosis has been widely reported, no consensus has been reached regarding the number of CpG sites that should be tested in *MGMT* promoter methylation. The main targets of the CpG methylation analysis should be clarified to establish the *MGMT* promoter methylation testing that can be generally utilized at all medical institutions. Accounting for the aforementioned considerations, we aimed to define which individual CpGs or CpG combinations could mainly affect the response to chemotherapy resulting from methylation in the statistical analysis focused on TVR. In pyrosequencing, methylation levels of combined CpG sites and MGMT protein expression correlated strongly. Everhard et al. studied methylation at 68 CpG sites and reported significant correlation of methylation levels of CpG 27, 32, 73, 75, 79, and 80 with *MGMT* mRNA suppression and association of combined CpG 32–33 and CpG 72–83 with *MGMT* mRNA suppression [6]. Chai et al. quantitatively analyzed methylation levels of CpG 72–82 using pyrosequencing and directly evaluated the predictive value of combined and individual CpG sites for *MGMT* mRNA expression. They revealed that CpG combinations with four or more consecutive CpGs within CpGs 72–82, including combined CpGs 76–79 and CpGs 74–78 used in commercial kits, were equally effective in predicting the *MGMT* mRNA expression and survival of patients with temozolomide-treated high-grade glioma [7], consistent with our findings. In an outcome-based study, Quillien et al. reported that the most powerful predictors of survival in patients with glioblastoma were CpG 84, CpG 89, and mean CpG 84–88 and that the mean CpG 74–78 was also strongly associated with the outcome of patients by a small margin [8]. Dahlrot et al. showed a correlation between *MGMT* promoter methylation in CpG 74–78 and MGMT protein suppression in glioblastoma, excluding non-tumor cells, using immunofluorescence staining [15]. *MGMT* promoter methylation at CpG sites upstream from CpG 82 is associated with survival of patients with glioblastoma [14,16,17]. In a meta-analysis of these studies, the Cochrane report concluded that a cutoff threshold of 9% for CpG sites 74–78 had the best predictive performance. Previous studies on the predictive performance of *MGMT* promoter methylation tested with pyrosequencing examined 4 to 16 CpG sites among CpG 74–89, and all of them included the first half of CpG 74–89, which correlated strongly with the prognosis [18].

In the present study, the most powerful predictors of temozolomide therapy efficacy in patients with glioblastoma were obtained from consecutive combinations including CpG 74–82 in exon 1, and combined CpG 78 and 79 was most strongly associated with temozolomide efficacy in terms of tumor volume change. In other words, methylation at these CpG sites in exon 1 could be associated with loss of MGMT activity. Therefore, combined CpG 76–79 and 74–78, targets of widely used Qiagen commercial kits, may help predict the loss of MGMT activity and efficacy of temozolomide therapy. These results are concordant with almost all aforementioned previous studies and are more practical because the cutoff values were determined by TVR rather than by survival, which may vary with the population or sample, as previously reported.

This study was limited by its retrospective design and relatively small population size, and further studies are required to validate these results. Intratumoral variations in the *MGMT* promoter methylation status might also have influenced these results. Therefore, future large-scale studies are warranted.

## 5. Conclusions

In conclusion, the methylation status of CpG 74–82 in exon 1 was most strongly associated with temozolomide efficacy, and the methylation analysis of these sites may help predict the loss of MGMT activity.

## Figures and Tables

**Figure 1 biomolecules-12-01379-f001:**
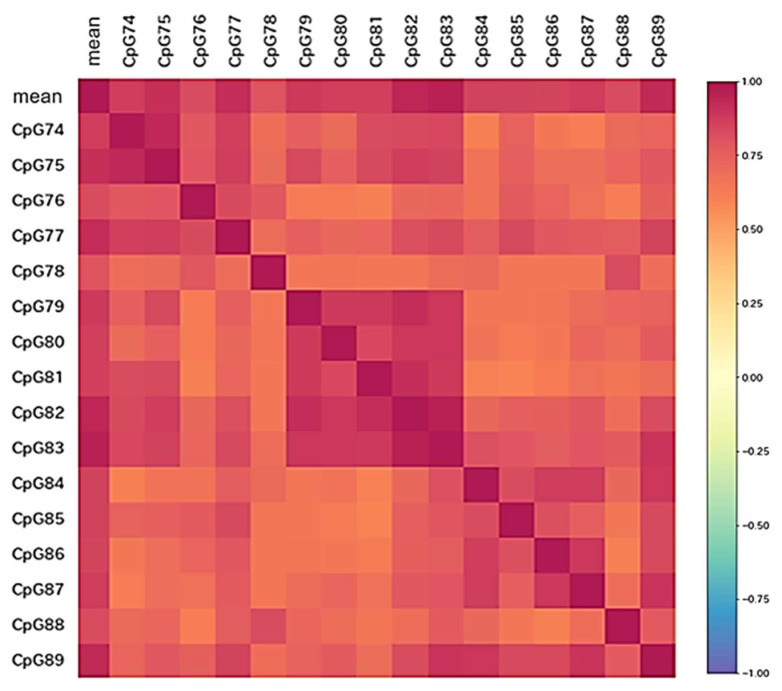
A heatmap shows correlation coefficients among methylation levels of all CpG sites and the average. CpG sites within the range from 79 to 83 shows a strong correlation.

**Figure 2 biomolecules-12-01379-f002:**
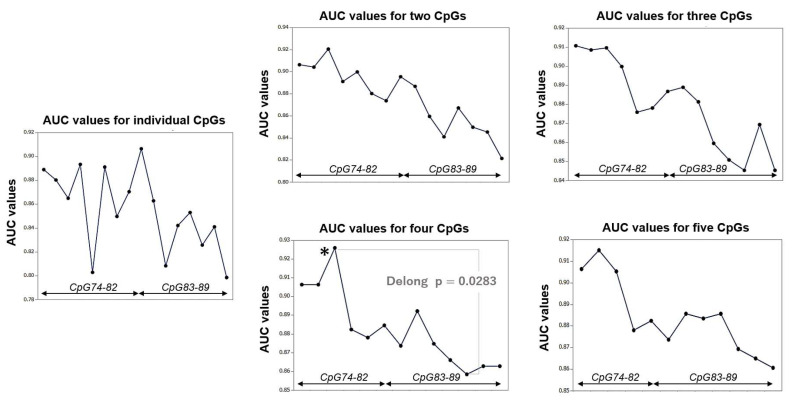
The line graphs show the AUCs at all individual and combined CpG sites, which indicate predictive performance for favorable response (CR or PR) to the adjuvant therapy. AUCs of consecutive CpG combinations were higher for CpG 74–82 in exon 1 than for CpG 83–89 in intron 1. *: highest value among all combinations.

**Table 1 biomolecules-12-01379-t001:** Baseline patient characteristics.

Variables		No. of Patients (%)
All patients		44
Age	Age≥65	23 (52.3%)
	Age<65	21 (47.7%)
Sex	Men	22 (50.0%)
	Women	22 (50.0%)
KPS	90	11 (25.0%)
	80	15 (34.1%)
	70	12 (27.3%)
	60	4 (9.1%)
	50	2 (4.6%)
Lesion	Single	38 (86.4%)
	Multiple	6 (13.6%)
Extent of removal	≥90%	19 (43.2%)
	<90%	25 (56.8%)
Ki-67 staining index	≥28.3% ^a^	22 (50.0%)
	<28.3%	22 (50.0%)
IDH1/2 mutation	not detected	44 (100.0%)
MGMTpm% CpG 74-82	≥9% ^b^	16 (36.4%)
	<9%	28 (63.6%)
MGMTpm% CpG 83-89	≥9%	20 (45.5%)
	<9%	24 (54.6%)

MGMTpm%: MGMT promoter methylation %. ^a^ the median Ki-67 staining index; ^b^ the cutoff value proposed in several previous studies.

**Table 2 biomolecules-12-01379-t002:** AUC value and LR+ calculated by ROC analysis.

Individual CpG	AUC	LR+	Combined CpGs	AUC	LR+	Combined CpGs	AUC	LR+	Combined CpGs	AUC	LR+	Combined CpGs	AUC	LR+
74	0.89	15.90	74, 75	0.91	15.90	74, 75, 76	0.91	15.90	74, 75, 76, 77	0.91	15.90	74, 75, 76, 77, 78	0.91	15.90
75	0.88	14.31	75, 76	0.90	14.31	75, 76, 77	0.91	15.90	75, 76, 77, 78	0.91	17.49	75, 76, 77, 78, 79	0.92	17.49
76	0.86	11.13	76, 77	0.92	15.90	76, 77, 78	0.91	19.08	76, 77, 78, 79	0.93	19.08	76, 77, 78, 79, 80	0.91	19.08
77	0.89	17.49	77, 78	0.89	17.49	77, 78, 79	0.90	20.67	77, 78, 79, 80	0.88	19.08	77, 78, 79, 80, 81	0.88	19.08
78	0.80	14.31	78, 79	0.90	22.26	78, 79, 80	0.88	19.08	78, 79, 80, 81	0.88	19.08	78, 79, 80, 81, 82	0.88	19.08
79	0.89	19.08	79, 80	0.88	17.49	79, 80, 81	0.88	19.08	79, 80, 81, 82	0.88	19.08	79, 80, 81, 82, 83	0.87	19.08
80	0.85	9.53	80, 81	0.87	17.49	80, 81, 82	0.89	19.08	80, 81, 82, 83	0.87	19.08	80, 81, 82, 83, 84	0.89	20.67
81	0.87	19.08	81, 82	0.90	19.08	81, 82, 83	0.89	19.08	81, 82, 83, 84	0.89	20.67	81, 82, 83, 84, 85	0.88	20.67
82	0.91	19.08	82, 83	0.89	19.08	82, 83, 84	0.88	20.67	82, 83, 84, 85	0.87	20.67	82, 83, 84, 85, 86	0.89	20.67
83	0.86	19.08	83, 84	0.86	19.08	83, 84, 85	0.86	17.49	83, 84, 85, 86	0.87	15.90	83, 84, 85, 86, 87	0.87	15.90
84	0.81	15.90	84, 85	0.84	14.31	84, 85, 86	0.85	11.13	84, 85, 86, 87	0.86	16.48	84, 85, 86, 87, 88	0.86	17.49
85	0.84	11.13	85, 86	0.87	12.72	85, 86, 87	0.85	14.31	85, 86, 87, 88	0.86	17.49	85, 86, 87, 88, 89	0.86	15.90
86	0.85	9.54	86, 87	0.85	7.95	86, 87, 88	0.87	17.49	86, 87, 88, 89	0.86	15.90	CpG74-82	0.91	19.08
87	0.83	11.13	87, 88	0.85	17.49	87, 88, 89	0.85	18.13				CpG83-89	0.87	19.08
88	0.84	14.31	88, 89	0.82	19.08							CpG74-89	0.88	20.67
89	0.80	15.90												

AUC: Area under curve, LR+: positive likelihood ratio. AUC and LR+, which indicate predictive performance for favorable response (CR or PR) to the adjuvant therapy, were calculated at individual and combined CpG sites.

## Data Availability

The data in this study are available from the corresponding author on reasonable request.

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
