# Peer review of "Volumetric Analysis of Glioblastoma for Determining Which CpG Sites Should Be Tested by Pyrosequencing to Predict Temozolomide Efficacy"

_biomolecules, 2022, doi:10.3390/biom12101379_

Round 1

Reviewer 1 Report

In this article, Hosoya et al. evaluated the correlation between the methylation at various CpG sites of MGMT to GBM patients prognosis following surgery and subsequent radiotherapy & temozolomide adjuvant. In particular, the authors evaluated the Tumor volume ratio (TVR) for receiver operating characteristic (ROC) analysis. Overall, the authors showed the methylation status of exon 1 CpG (especially CpG 76-79) to be most correlated to TVR reduction. 

As the authors mentioned, CpG methylation status of MGMT is widely adopted to predict the prognosis of GBM patients. This study strengthens this notion further while highlighting which CpG site requires careful examination through Japan clinical patient data. 

Overall, I find the article to be clear and concise, greatly suitable to the readers of Biomolecules. However, there remain several points which require clarification and revision from the authors:

- Manuscript guideline sentences in page 2 line 89-99 should be removed

- The authors mentioned 60Gy localized radiotherapy alongside adjuvant temozolomide therapy. However, the authors did not provide when and how frequent such therapy was given. Please provide such description in the methodology, as ensuring standardization can affect the treatment efficacy observed.

- Please define KPS

- Scaling legend for the heatmap in Figure 1 would be helpful

- Considering the relatively poor AUC & LR+ ratio for CpG 78 (as compared to CpG 77 & 79), I wonder the reason how combined CpG 78-79 can show most optimal LR+. Do the authors have explanation for such observation?

Author Response

Dear Reviewer 1,

We are grateful for your consideration of our manuscript entitled “Volumetric analysis of glioblastoma for determining which CpG sites should be tested by pyrosequencing to predict temozolomide efficacy” and appreciate the time and effort you and the reviewers have dedicated to providing insightful feedback, which has greatly helped us improve our manuscript.

In reply to the comments, and especially to the comment from reviewer 2 about English language revision, our manuscript went through the English amendment process by Editage.com and has been substantially revised. We are aware that Biomolecules instructed us to use “track change” function of Word, but since there are so many corrections of English language which is difficult to see, we are submitting the revised version with the English amendment part finalized and the response to the reviewers’ comments part highlighted in yellow.

Please evaluate our point-by-point responses to the comments below. Blue color indicates the reviewers' comments.

Reviewer 1

- Manuscript guideline sentences in page 2 line 89-99 should be removed

→Thank you for your thoughtful comments. These sentences were removed.

- The authors mentioned 60Gy localized radiotherapy alongside adjuvant temozolomide therapy. However, the authors did not provide when and how frequent such therapy was given. Please provide such description in the methodology, as ensuring standardization can affect the treatment efficacy observed.

→Thank you for your thoughtful comments. Local radiotherapy and chemotherapy were simultaneously administered within a month of surgery as same as the golden standard therapy called Stupp regimen. We added the wording in the Materials and Methods part (line 88-90) as follows; ”Local radiotherapy (60 Gy in 30 fractions) and chemotherapy were simultaneously administered within a month of surgery for all patients.”

- Please define KPS

→Thank you for your thoughtful comments. In the field of neuro-oncology, we conventionally assess patients’ performance by using Karnofsky Performance Scale (KPS) [1] instead of ECOG Performance Status (PS). We added the wording in the Results part (line 149). In addition, more details of KPS were listed in Table1.

[1] Karnofsky DA, Ablemann WH, et al.: The use of nitrogen mustards in the palliative treatment of carcinoma. Cancer. 1948;1:634-656.

- Scaling legend for the heatmap in Figure 1 would be helpful

→Thank you for your thoughtful comments. We added the scaling legend in Figure1.

- Considering the relatively poor AUC & LR+ ratio for CpG 78 (as compared to CpG 77 & 79), I wonder the reason how combined CpG 78-79 can show most optimal LR+. Do the authors have explanation for such observation?

→Thank you for your thoughtful comments. Malley et al [2] showed in a series of experiments in which each CpG was replaced by TpG either individually or in combination that, in general, replacing multiple CpGs had greater effect of downregulating promoter activity than single CpGs. Although a combination of replacing CpG 78 and CpG 79 was not tested in this paper, it is conceivable that a combination of CpG 78 and CpG 79 methylation may more efficiently suppress MGMT expression, possibly by altering the binding efficacy of a transcription factor in the MGMT promoter.

[2] Malley DS, Hamoudi RA, et al.: A distinct region of the MGMT CpG island critical for transcriptional regulation is preferentially methylated in glioblastoma cells and xenografts. Acta Neuropathol. 2011;121(5):651-61.

Again, we appreciate the opportunity to improve our manuscript with your valuable comments and queries.

Sincerely,

Masamichi Takahashi, MD, PhD

*Corresponding author

Assistant Chief, Department of Neurosurgery and Neuro-Oncology, National Cancer Center Hospital,

5-1-1, Tsukiji, Chuo-ku, Tokyo, 104-0045, Japan

Phone: +81-3-3542-2511

FAX: +81-3-3542-2551

Reviewer 2 Report

In the current study, Hosoya et al. describe the effect of individual (or combination) CpG island methylation on response to chemo-radiation therapy in IDH-wildtype glioblastoma. I appreciate their use of TVR in assessing effects of temezolamide, rather than commonly utilized outcomes such as overall survival or progression free survival. Overall, I think the manuscript provides interesting and potentially useful data in terms of analyzing MGMT promotor methylation in GBM patients; however, some issues need to be addressed:

- How was IDH-wildtype status confirmed? It is mentioned that pyrosequencing was performed for IDH1/2 in the methods but these results are not presented in the Results section and should be included when describing the cohort. 

- How would the authors define whether or not the MGMT promotor is considered "methylated" vs "not methylated". This binary output is useful for treating physicians when making decisions about whether TMZ should be utilized. Similarly, how do other labs define whether or not the MGMT promoter is methylated, and how does these results compare with the current study?

- Formatting and English language revisions need to be incorporated to improve the readability and professionalism of the manuscript. There are numerous instances of subject-verb disagreement, incorrect verb tense, inappropriate use of indefinite articles, and awkward transitions. Similarly, "AUC" is not defined in the abstract. The term "et al" should more correctly be "et al." (with a period). There is also a large portion of text in the last paragraph of the introduction that does not belong. 

- The introduction is currently too long and should be condensed.

Author Response

Dear Reviewer 2,

We are grateful for your consideration of our manuscript entitled “Volumetric analysis of glioblastoma for determining which CpG sites should be tested by pyrosequencing to predict temozolomide efficacy” and appreciate the time and effort you and the reviewers have dedicated to providing insightful feedback, which has greatly helped us improve our manuscript.

In reply to the comments, and especially to the comment from reviewer 2 about English language revision, our manuscript went through the English amendment process by Editage.com and has been substantially revised. We are aware that Biomolecules instructed us to use “track change” function of Word, but since there are so many corrections of English language which is difficult to see, we are submitting the revised version with the English amendment part finalized and the response to the reviewers’ comments part highlighted in yellow.

Please evaluate our point-by-point responses to the comments below. Blue color indicates the reviewers' comments.

Reviewer 2

- How was IDH-wildtype status confirmed? It is mentioned that pyrosequencing was performed for IDH1/2 in the methods but these results are not presented in the Results section and should be included when describing the cohort.

→Thank you for your thoughtful comments. By using pyrosequencing method [3], IDH1/2 mutated gliomas were excluded from the cohort, so only cases without IDH1/2 mutation, that is equivalent to “Glioblastoma, IDH-wildtype” in WHO 2021 classification, were eligible for our study. These results were added in Table1 and described in Results section as “IDH-wildtype was confirmed in all the patients” (line 152-153), and the Legend of Table 1 (line 161-162).

[3] Arita H, Narita Y, Matsushita Y, Fukushima S, Yoshida A, Takami H, Miyakita Y, Ohno M, Shibui S, Ichimura K (2015) Development of a robust and sensitive pyrosequencing assay for the detection of IDH1/2 mutations in gliomas. Brain Tumor Pathol 32: 22-30 doi:10.1007/s10014-014-0186-0

- How would the authors define whether or not the MGMT promotor is considered "methylated" vs "not methylated". This binary output is useful for treating physicians when making decisions about whether TMZ should be utilized. Similarly, how do other labs define whether or not the MGMT promoter is methylated, and how does these results compare with the current study?

→Thank you for your thoughtful comments. Recently, previous meta-analysis studies, the Cochrane report concluded that a cutoff threshold of 9% for CpG sites 74-78 had the best predictive performance for survival [4]. On the other hand, in our institution, the cutoff point of MGMT gene promoter methylation is empirically set at 16% for CpG sites 74-89 based on our data. Since the MGMT promoter methylation status cutoff values may differ depending on the median OS of the cohorts used for stratification, definite cutoff value that discriminates MGMT promoter methylation status has not yet been determined. For the reason mentioned above, we have focused on the Tumor Volume Ratio (TVR) analysis in our studies. Although the MGMT gene promoter methylation status is mostly determined by using Methylation Specific PCR (MSP) method, the pyrosequencing method might have relatively better specificity and sensitivity than the MSP method according to the previous reports [4,5].

[4] Brandner S, McAleenan A, Kelly C, Spiga F, Cheng HY, Dawson S, Schmidt L, Faulkner CL, Wragg C, Jefferies S, Higgins JPT, Kurian KM (2021) MGMT promoter methylation testing to predict overall survival in people with glioblastoma treated with temozolomide: a comprehensive meta-analysis based on a Cochrane Systematic Review. Neuro Oncol 23: 1457-1469 doi:10.1093/neuonc/noab105

[5] Lattanzio L, Borgognone M, Mocellini C, Giordano F, Favata E, Fasano G, Vivenza D, Monteverde M, Tonissi F, Ghiglia A, Fillini C, Bernucci C, Merlano M, Lo Nigro C (2015) MGMT promoter methylation and glioblastoma: a comparison of analytical methods and of tumor specimens. Int J Biol Markers 30:e208-216. https://doi.org/10.5301/jbm.5000126

- Formatting and English language revisions need to be incorporated to improve the readability and professionalism of the manuscript. There are numerous instances of subject-verb disagreement, incorrect verb tense, inappropriate use of indefinite articles, and awkward transitions. Similarly, "AUC" is not defined in the abstract. The term "et al" should more correctly be "et al." (with a period). There is also a large portion of text in the last paragraph of the introduction that does not belong.

→We appreciate your thoughtful comments. We revised and amended our manuscript according to your advice with the support of native English editor from Editage.com. Since the manuscript is significantly amended, the English revision part is already finalized and the responses to the reviewers’ comments are highlighted in yellow. We also added wording regarding English revision in Acknowledgement.

- The introduction is currently too long and should be condensed.

→Thank you for your thoughtful comments. These sentences in the introduction part were condensed.

Again, we appreciate the opportunity to improve our manuscript with your valuable comments and queries.

Sincerely,

Masamichi Takahashi, MD, PhD

*Corresponding author

Assistant Chief, Department of Neurosurgery and Neuro-Oncology, National Cancer Center Hospital,

5-1-1, Tsukiji, Chuo-ku, Tokyo, 104-0045, Japan

Phone: +81-3-3542-2511

FAX: +81-3-3542-2551
